# Poor Self-Rated Health and Associated Factors among Older Persons in Malaysia: A Population-Based Study

**DOI:** 10.3390/ijerph20054342

**Published:** 2023-02-28

**Authors:** Norhafizah Sahril, Yee Mang Chan, Ying Ying Chan, Noor Ani Ahmad, Mohd Shaiful Azlan Kassim, Nik Adilah Shahein, Muhammad Solihin Rezali, Mohamad Aznuddin Abd Razak, Fatin Athirah Tahir, Nor’ain Ab Wahab, Norliza Shamsuddin, Muhammad Azri Adam Adnan, Muhamad Khairul Nazrin Khalil, Siaw Hun Liew

**Affiliations:** Institute for Public Health, National Institutes of Health, Ministry of Health Malaysia, Shah Alam 40170, Malaysia

**Keywords:** self-rated health, older persons, National Health and Morbidity Survey, Malaysia

## Abstract

Objective: This study aims to determine the prevalence of poor self-rated health (SRH) in Malaysia and its association with sociodemographic characteristics, lifestyle behavior, chronic diseases, depression, and functional limitations in activities of daily living (ADLs) among older persons. Design: Cross-sectional. Setting, participants, and outcome measures: We used data from the National Health and Morbidity Survey 2018, a nationwide community-based study. This study was conducted using a two-stage stratified cluster sampling design. Older persons were defined as persons aged 60 years and above. SRH was assessed using the question “How do you rate your general health?” and the answers were “very good”, “good”, “moderate”, “not good”, and “very bad”. SRH was then grouped into two categories; “Good” (very good and good) and “Poor” (moderate, not good, and very bad). Descriptive and logistic regression analyses were conducted using SPSS version 25.0. Results: The prevalence of poor SRH among older persons was 32.6%. Poor SRH was significantly related to physical inactivity, depression, and limitations in activities of daily living (ADLs). Multiple logistic regression revealed that poor SRH was positively associated with those who had depression (aOR 2.92, 95% CI:2.01,4.24), limitations in ADLs (aOR 1.82, 95% CI: 1.31, 2.54), low individual income (aOR 1.66, 95% CI:1.22, 2.26), physical inactivity (aOR 1.40, 95% CI:1.08, 1.82), and hypertension (aOR 1.23, 95% CI:1.02, 1.49). Conclusions: Older persons with depression, limitations in ADLs, low income, physical inactivity, and hypertension were significantly associated with poor SRH. These findings provide information to aid health personnel and policymakers in the development and implementation of health promotion and disease prevention programs, as well as adequate evidence in planning different levels of care for the older population.

## 1. Introduction

Self-rated health (SRH) is a subjective measurement that integrates the biological, social, mental, and functional aspects of an individual [1]. Even though it may be subjective to individuals, the World Health Organization (WHO) recommends SRH as one of the indicators for health monitoring because it has been shown to be a reliable predictor and measure of health outcomes such as morbidity, mortality, functional difficulties and chronic diseases in various populations [2,3,4,5,6]. SRH was developed by John Ware and Cathy Sherbourne in 1992 to capture how healthy people think they are [7]. Typically, SRH is measured in a single item that asked respondents to rate their current health status on a five-point scale ranging from “very good” to “very poor” [8]. Previous research has found that demographic factors (such as age, gender, income, and education) [9], as well as lifestyle, psychological well-being [10], social and emotional support [11], chronic diseases, and functional factors (such as physical and basic daily activities) [12,13] all have an impact on SRH among older persons.

A comparison study of self-rated health among older persons in China and the United States (US) found that Chinese respondents were much more likely to rate their health as poor than US respondents. In both countries, older persons with limitations in activities of daily living (ADLs), poor mental health, and chronic health conditions were reported to have had lower self-rated health [14]. In Ghana, older persons who had never worked in their lives, resided in rural areas, and had functional difficulties were found to be more likely to report poor SRH [15]. Poor SRH was reported to be significantly associated with low income, older age, not working, poor functional capacity, and depression in both older men and women in a Brazilian study [16]. According to a study by Yang H et al. (2021) of 6551 older people in Zhongshan, China, those who have a chronic disease, mental health symptoms, and poor social relationships were significantly more likely to have poor self-rated health [17]. Meanwhile, a Taiwanese study of 312 urban community-dwelling older persons aged 65 to 90 years found that poor SRH was associated with high income, abstinence from alcohol, being physically inactive, feeling hopeless, and having difficulties in activities of daily living [11]. A 2020 study published in the Journal of Epidemiology and Global Health found that among elderly individuals in Indonesia, 46.4% reported poor self-rated health (SRH). The study revealed that poor SRH was strongly linked with factors such as advanced age, female gender, lower educational attainment, lower income, and chronic illnesses such as hypertension, diabetes, and heart disease [18]. Similarly, the prevalence of poor SRH among the elderly in India was found to be 56.0%, with similar associations with age, gender, education, income, and chronic diseases. These findings were also reported in the same study [19].

In Malaysia, there is limited data to assess SRH. Two studies focused solely on exploring the sociodemographic characteristics of SRH among older persons [20,21]. Another study by Teh et al. (2014) found that sociodemographic characteristics, socioeconomic factors, physical limitations, non-communicable diseases (NCDs), and lifestyle factors were significant predictors of SRH [22]. However, the data used in this study came from the Malaysian Population and Family Survey, which was conducted in 2004. Hence, by using the most recent Malaysian older population data from the National Health and Morbidity Survey 2018 (NHMS 2018), we aimed to determine the prevalence of poor SRH and its relations to sociodemographic characteristics, lifestyle behavior, chronic diseases, depression, and activities of daily living (ADLs) among older persons in this country.

Variables such as sex, locality, education level, employment status, marital status, individual monthly income, living arrangements, lifestyle behaviors, chronic diseases, depression, and activities of daily living are commonly used in studies related to health and well-being. These variables are chosen because they are known to be associated with poor health outcomes, and can provide important insights into the factors that contribute to poor self-rated health. For example, women tend to report worse self-rated health compared to men, and individuals with lower levels of education and income tend to have worse health outcomes than those with higher levels. Similarly, living in rural areas and being unemployed or underemployed can also impact health outcomes. Being female, older age, lower education level, lower income, and not being married were associated with poor SRH [22]. Lifestyle behaviors such as smoking, physical activity, and consumption of fruits and vegetables are important predictors of health outcomes. Being physically inactive and having a poor diet were associated with poor SRH [22]. According to a study by Kandiah and Japaraj (2019), older smokers were more likely to report poor self-rated health compared to non-smokers, even after controlling for demographic, socioeconomic, and health-related factors. The study also found that older persons who quit smoking had better self-rated health compared to those who continued smoking [23]. Chronic diseases such as diabetes, hypertension, and hypercholesterolemia are common health problems among older adults in Malaysia and are known to impact self-rated health. Several chronic conditions, including hypertension and diabetes, were associated with poor SRH [22]. Depression is also a common mental health problem among older persons and can contribute to poor self-rated health [22]. Finally, activities of daily living (ADLs) are essential for maintaining independence and quality of life and are often used as a measure of functional status among older adults. Impairments in ADLs can indicate poor health and reduced ability to perform daily tasks. Having limitations in activities of daily living (ADLs) was strongly associated with poor SRH [22]. By considering these variables in a study, researchers can better understand the factors that contribute to poor self-rated health among older persons in Malaysia and develop targeted interventions to improve health outcomes.

Examining the link between self-rated health and socioeconomic factors is of utmost importance for multiple reasons. By exploring the association between self-rated health and socioeconomic determinants, we can identify the factors contributing to these health disparities and create strategies to mitigate them. Additionally, understanding the correlation between self-rated health and socioeconomic determinants is crucial in identifying the populations that are most at risk of poor health outcomes, and developing targeted interventions to promote health equity. Furthermore, as socioeconomic determinants are often influenced by policies and programs, this link can aid in the development and planning of policies and programs that can improve health outcomes. Finally, recognizing the relationship between self-rated health and socioeconomic determinants can help identify obstacles to healthcare access and guide efforts to improve access for disadvantaged populations. Overall, this analysis is vital for promoting health equity and enhancing health outcomes for all individuals.

Malaysia has an aging population, with the proportion of older persons (aged 60 years and above) projected to increase from 9% in 2020 to 14.5% in 2040. In summary, the Malaysia dataset on SRH would be interesting because it will provide insights into the health status and well-being of older persons, the relationship between self-rated health and socioeconomic determinants, the Malaysian health care system, and cross-cultural comparisons of SRH. These insights can inform policies and interventions that promote healthy aging and improve the health and well-being of all Malaysians.

## 2. Methodology

### 2.1. Study Design and Sampling Method

This study used data from the National Health and Morbidity Survey 2018 (NHMS 2018), which focused on the Malaysian population aged 60 years and above. It was a nationwide, population-based cross-sectional survey with a two-staged stratified cluster sampling design. The primary stratum was made up of all thirteen states and three federal territories, and the secondary stratum is comprised of both urban and rural areas to ensure that the sample size was representative of the population at the national level. Using a single proportion formula and previous estimates from the literature, the sample size was calculated for each specific objective based on the identified scopes. The estimated sample size (3542) was calculated using the largest sample size based on the prevalence of living alone. This sample size was chosen to ensure that the sample in this study covers all of the NHMS 2018 objectives [24]. The selection of samples was carried out by the Department of Statistics Malaysia with sixty enumeration blocks in the urban area, and fifty enumeration blocks in the rural area. All older persons within the selected living quarters were included in this survey. This study included a total of 3997 older persons.

### 2.2. Data Collection Process and Tools

Data collection for the NHMS 2018 was carried out from July to September 2018. A bilingual (Malay and English) structured questionnaire was designed, pre-tested, and piloted before the actual survey. Face-to-face interviews using a mobile tablet device were carried out by trained interviewers. Written consent was obtained from all eligible respondents before the interviews. The study protocol was approved by the Medical Research and Ethics Committee (MREC), Ministry of Health Malaysia (NMRR-17-2655-39047). A detailed elaboration of the NHMS 2018 survey’s methodology can be found in [24].

### 2.3. Patient and Public Involvement

Respondents were not involved in the design, conducting, reporting, or dissemination plans of our study.

## 3. Measures

### 3.1. Dependent Variable

Self-rated health was measured based on the question: “How do you rate your general health?” and the answers were “very good”, “good”, “moderate”, “not good”, and “very bad”. For the purpose of the analysis, the responses of self-rated health were collapsed into “Good” (very good and good) and “Poor” (moderate, not good, and very bad). In the logistic regression analysis, “Good” SRH was used as the reference category.

### 3.2. Independent Variables

#### 3.2.1. Sociodemographic Variables

The following sociodemographic variables were included in the analysis: sex (male, female), locality (urban, rural), education level (no formal education, primary, secondary, tertiary), employment status (employed, unemployed), marital status (single, married), individual monthly income (<MYR 1000, MYR 1000–MYR 1999, ≥MYR 2000), and living arrangements (living alone, living with a spouse, partner, or anyone else in the house).

#### 3.2.2. Chronic Diseases

The presence of chronic diseases was based on self-reporting of being medically diagnosed (by a doctor or assistant medical officer) with diabetes mellitus, hypertension, or hypercholesterolemia.

#### 3.2.3. Lifestyle Behavior

For smoking status, an individual who was currently using any smoked tobacco products was considered a current smoker. The Global Physical Activity Questionnaire (GPAQ) [25] was used to measure self-reported physical activity. The level of physical activity is calculated in terms of metabolic energy expanded (MET) minutes (METs). Respondents were classified as physically active if they achieved at least 600 METs per week of vigorous-intensity, moderate-intensity, or walking activities. Those respondents who achieved less than 600 METs per week were classified as inactive. Consumption of fruits and vegetables was measured based on four questions: (i) “In a typical week, how many days do you eat fruits?”; (ii) “Usually on the day you eat fruits, how many servings of fruits did you eat in a day?”; (iii) “In a typical week, how many days do you eat vegetables?;” (iv) “Usually on the day you eat vegetables, how many servings of vegetables did you eat in a day?”. A codebook containing photos of food was used to assist respondents in recalling the serving size of the fruits and vegetables they had consumed. The answers to these four questions were used to calculate the total average daily consumption of fruits and vegetables. Consumption of fruits and vegetables was divided into two groups based on World Health Organization (WHO) recommendations: adequate consumption (≥5 servings per day) and inadequate consumption (<5 servings per day) [26].

#### 3.2.4. Mental Health Problems

The previously validated Geriatric Depression Scale-14 was used to assess mental health problems, which included depressive symptoms, and a cut-off point of >6 (out of 14) indicates a positive screening result [27].

#### 3.2.5. Functional status

The Barthel Index is an ordinal scale used to measure performance in activities of daily living (ADLs). The Barthel includes 10 functional elements of self-care such as feeding, personal toileting, bathing, dressing, and undressing, getting on and off a toilet, controlling the bladder, controlling the bowel, moving from a wheelchair to bed and returning, walking on a level surface (or propelling a wheelchair if unable to walk), and ascending and descending stairs. The requirement of assistance due to an inability to independently perform one or more ADLs indicates functional limitations and the need for supportive services. The index was divided into two categories: a total maximum score of 20 was categorized as an absence of functional limitation and a total score below 20 was categorized as the presence of functional limitation [28].

## 4. Analysis

Data analysis was conducted using SPSS Statistics 25.0 for Windows (IBM Corp., Armonk, NY, USA) taking into account the complex survey design. First, the overall prevalence of SRH was determined. Associations of the measured factors with poor SRH were tested using chi-square tests. The dependent variable was dichotomous in nature; thus, the logistic regression analysis was used in this study to calculate the crude odds ratio (OR) as a measure of association. The adjusted ORs, along with their respective 95% confidence intervals (CIs), were calculated by including all selected independent variables in the final multivariable logistic regression model. *p*-values less than 0.05 were considered in assessing the significant association.

## 5. Results

Table 1 shows the sociodemographic profile of the respondents. This study includes a total of 3977 older people aged 60 and above. There was a slightly higher proportion of women (52.9%), with 57.5% residing in rural areas, 48.8% having at least a primary education level, and 63.3% having at least MYR 1000.00 individual monthly income. The majority of older persons were not living alone (92.6%); 73.6% were unemployed and 66.0% were married.

Table 2 shows the prevalence of poor SRH by sociodemographic characteristics, lifestyle behavior, chronic diseases, depression, and limitations in ADLs. The prevalence of poor self-rated health (SRH) among older persons was 32.6% (95% CI: 28.83, 36.68). Poor SRH was associated with self-reported diabetes, hypertension, hypercholesterolemia, depressive symptoms, physically inactive, and limitations in ADLs. Other possible factors included those respondents that are less educated, single, unemployed, and from a low economic background (*p* < 0.001).

Table 3 shows the results of univariate and multivariate logistic regression analysis for poor SRH by sociodemographic characteristics, lifestyle behavior, chronic diseases, depression, and limitations in ADLs. Further multivariate analysis using logistic regression revealed that older persons with depression had nearly three times higher odds of having poor SRH (aOR 2.92, 95% (CI: 2.01, 4.24, *p* < 0.001)), while those with limitations in ADLs had nearly double the odds of having poor SRH (aOR 1.82, 95% (CI: 1.31, 2.54, *p* < 0.001)). Other risk factors for poor SRH among older persons were low individual income (aOR 1.66, (95% CI: 1.22, 2.26, *p* = 0.001)), physically inactive (aOR 1.40, (95% CI: 1.08, 1.82, *p* = 0.011)), and self-reported medically diagnosed hypertension (aOR 1.23, (95% CI: 1.02, 1.49, *p* = 0.032)).

## 6. Discussion

The prevalence of poor self-rated health (SRH) among Malaysia’s older population was found to be lower at 32.6% in comparison to China, Indonesia, and India, as reported in references [17,18,19]. However, the prevalence was higher in comparison to Ghana and Brazil as stated in references [15,16]. This difference in prevalence may be attributed to various factors, such as the study population’s demographic characteristics, including age, gender, and ethnic group, and the methodology used, such as the type of rating scale and the wording of response categories. Additionally, the differences in health status among older persons may be influenced by cultural, geographical, socioeconomic, and socio-political factors. These factors can affect an individual’s perception of their own health, and, therefore, may contribute to differences in SRH prevalence across different countries. For instance, differences in cultural beliefs and values regarding health and illness may impact the way older persons perceive and rate their health status. Similarly, variations in access to healthcare services and resources, such as medication and medical facilities, can influence the likelihood of chronic illnesses, which in turn can affect SRH. Socioeconomic factors, such as income and education levels, may also impact an individual’s perception of their own health, as those with lower socioeconomic status may experience higher levels of stress, which can negatively impact physical and mental health. Furthermore, socio-political factors such as government policies, healthcare systems, and public health programs can also influence the health status of older individuals. For example, the availability and accessibility of preventive health care measures, such as vaccinations and health education programs, may influence the incidence of chronic diseases, and in turn, the prevalence of poor SRH among older persons. In conclusion, the difference in the prevalence of poor SRH among older persons in Malaysia, China, Indonesia, India, Ghana, and Brazil can be attributed to a multitude of factors, including demographic characteristics, methodological differences, and cultural, geographical, socioeconomic, and socio-political factors. Understanding these factors is crucial in designing effective health policies and programs that can improve the health and well-being of older persons in different countries.

The current study’s findings indicate a higher prevalence of SRH among older persons in Malaysia than previously reported in studies, with rates of 17.0% and 17.2% in references [20,21], respectively. These differences in prevalence may be attributed to variations in the study population and measurement instruments used. For instance, the previous studies may have had different inclusion and exclusion criteria or used different measures to assess SRH. Moreover, differences in the prevalence rates may also be influenced by short-term fluctuations in health or illness resulting from cyclical changes in well-being. These fluctuations may result from various factors, such as seasonal changes or acute illnesses, which can affect an individual’s perception of their health status. Additionally, changes in an individual’s social, economic, or physical environment can also impact their well-being and, consequently, their SRH ratings. It is also worth noting that the current study’s higher prevalence of poor SRH among older persons in Malaysia could reflect a genuine increase in health problems among the aging population. With advancing age, individuals may experience age-related physiological and functional changes that can contribute to a decline in health status. Moreover, older individuals may have a higher likelihood of developing chronic illnesses, such as hypertension, diabetes, or heart disease, which can impact their SRH ratings. Overall, the differences in the prevalence rates of poor SRH among older persons in Malaysia reported in the current study and previous studies could be explained by various factors, including differences in the study population, measurement instruments, short-term fluctuations in health, and genuine changes in health status. Future research should consider these factors when conducting SRH assessments to ensure accurate and reliable estimates of SRH prevalence rates among older persons in Malaysia.

We discovered that chronic disease, such as hypertension, was a determinant of SRH, which is consistent with other previous studies [17,18,19,29,30]. Hypertension, also known as high blood pressure, is a common condition among older persons, and it can have a significant impact on their overall health. In older persons with hypertension, SRH may be related to a variety of factors. For example, uncontrolled blood pressure can cause a variety of health problems, such as heart disease, stroke, kidney disease, and vision loss. These conditions can have a significant impact on an individual’s physical health and well-being and can contribute to a negative perception of their overall health status. In addition to physical health, psychological factors can also play a role in self-rated health. For example, older persons with hypertension may experience anxiety or depression related to their condition, which can negatively impact their self-rated health. Social factors, such as isolation or lack of support, can also contribute to poor self-rated health. To improve self-rated health in older persons with hypertension, it is important to focus on controlling blood pressure and managing any related health conditions. This may involve medication, lifestyle changes, and regular medical monitoring. In addition, addressing any psychological or social factors that may be contributing to poor self-rated health can also be beneficial. This may involve counseling, social support, or other interventions to improve overall well-being.

Previous research has indicated a possible significant association between diabetes and poor self-rated health in the older population [14,17,18,19,22,30]. Because diabetes can have a severe impact on an individual’s health, it was initially assumed that it would have a more adverse effect on self-rated health (SRH) compared to other chronic conditions, such as hypertension. However, recent studies have surprisingly found that the impact of diabetes on SRH is not significantly worse than that of hypertension. Although diabetes is linked to various comorbidities and complications, such as nerve damage, kidney disease, and cardiovascular disease, hypertension can also result in severe health issues, including heart attacks, strokes, and kidney damage. The study’s outcomes suggest that the effect of diabetes on an individual’s self-rated health is not as severe as previously thought. This highlights the importance of considering other factors that can affect an individual’s perception of their health status, such as the severity of the illness, age, and the presence of comorbidities. To conclude, while diabetes is a severe medical condition that necessitates careful management, it is not necessarily more damaging to an individual’s self-rated health than other chronic illnesses such as hypertension. The study’s findings emphasize the need for personalized approaches to managing chronic diseases and accounting for the individual’s overall perception of their health.

Having limitations in carrying out activities of daily living (ADLs) is an important factor that is linked to lower self-rated health among older persons in Malaysia [30], as well as in other countries [1,11,14,15,31]. Our study findings support this association, indicating that respondents who reported difficulties with ADLs had a significantly higher likelihood of reporting poor self-rated health. This suggests that ADLs are crucial measures of an individual’s functional status and ability to perform basic daily tasks, and if an older person experiences challenges in performing these tasks, it can have a negative impact on their overall health and well-being, ultimately increasing the likelihood of reporting poor self-rated health.

In our study, depression was another important factor influencing poor SRH, which is in line with previous studies [16,17,22]. There are several reasons why older persons with depression may be more likely to report poor self-rated health. Depression is a mood disorder that can affect many aspects of a person’s life, including physical health, social relationships, and overall quality of life. It can also contribute to the development or exacerbation of other chronic health conditions, such as cardiovascular disease, diabetes, and arthritis, which can further impact an individual’s ability to perform daily activities and maintain a good quality of life. Depression can also lead to changes in behavior and lifestyle, such as decreased physical activity, poor nutrition, and sleep disturbances, which can negatively affect physical health and increase the risk of developing chronic health problems. These changes can also lead to a loss of independence, reduced social interaction, and feelings of isolation, which can contribute to poor self-rated health and a decline in overall well-being. Additionally, depression can also make it difficult for older adults to manage their health conditions effectively, leading to poor self-care, medication non-adherence, and decreased engagement with healthcare providers, which can further worsen physical health outcomes. Overall, depression in older adults can have a significant impact on physical and mental health outcomes and can contribute to poor self-rated health. By identifying and addressing depression in older adults, healthcare providers may be able to help improve both mental and physical health outcomes and promote healthy aging. The findings indicate that mental health services for older persons should be improved and monitored. Healthcare providers, public health officials, and policymakers must take action to ensure that individuals suffering from depression receive the appropriate treatment.

Our finding reported older persons with low individual incomes had a significant impact on their self-rated poor health. The outcome is consistent with the results from a previous study conducted by Teh et al. (2014), which found a significant association between lower income and self-rated poor health among older persons [22]. The reason for these findings could be that low-income older people face greater budget constraints in accessing good resources such as healthy food, a home or shelter, and medical care, and are thus more likely to suffer from poor health. The rapid growth of Malaysia’s economy, coupled with rising living costs, is thought to have an influence, particularly among these lower socioeconomic people. The Eleventh Malaysian Plan’s targets and strategies to assist raise the economic and income level of this community have been acknowledged and should benefit them.

Older persons who were physically inactive were more likely to report poor SRH in this study. It is generally well-established that older persons who are physically inactive are more likely to report poor self-rated health [11,22]. This is because physical activity is an important factor in maintaining overall health and well-being. Regular physical activity has numerous health benefits for older persons, such as improving cardiovascular health, strengthening muscles and bones, reducing the risk of falls, and improving mental health. On the other hand, physical inactivity can contribute to a range of health problems, such as obesity, cardiovascular disease, and depression. In addition to the physical health benefits, regular physical activity can also have positive impacts on an individual’s perception of their own health. Engaging in physical activity can lead to feelings of accomplishment, increased confidence, and a greater sense of well-being, all of which can contribute to a more positive self-rated health status. Therefore, it is important to encourage older individuals to engage in regular physical activity as part of their overall health and wellness plan. This may involve structured exercise programs, such as tai chi, yoga, or strength training, or simply encouraging regular walking or other forms of moderate physical activity. By promoting regular physical activity, we can help improve the physical and psychological well-being of older persons and ultimately improve their self-rated health.

The use of a large population-based sample is one of this study’s strengths. Furthermore, this study provides an update on SRH status and its determinants among older persons in Malaysia. However, there are several limitations to the study that should be mentioned. First, because the study is based on cross-sectional data, causal relationships between SRH and the variables studied should be interpreted with caution. Another limitation is that the chronic diseases were self-reported, which may have resulted in information bias.

## 7. Conclusions

The present study reported that depression, limitations in ADL, low income, physical inactivity, and hypertension are major factors influencing poor SRH. Having knowledge of the self-rated health (SRH) status of older individuals and the usual determinants of SRH can aid healthcare providers, public health officials, policymakers, and public health sectors in concentrating their efforts towards enhancing the health and functioning of the older population.

## Figures and Tables

**Table 1 ijerph-20-04342-t001:** Sociodemographic profile of the respondents (n = 3977).

Variables	Unweighted Count	Percentage (%)
Gender		
Male	1872	47.1
Female	2105	52.9
Locality		
Urban	1689	42.5
Rural	2288	57.5
Education Level		
Non-formal	806	20.3
Primary	1939	48.8
Secondary	967	24.3
Tertiary	265	6.7
Employment Status		
Employed	1050	26.4
Unemployed	2927	73.6
Marital Status		
Single	1350	33.9
Married	2624	66.0
Individual Monthly Income		
<MYR 1000	2519	63.3
MYR 1000–MYR 1999	845	21.2
≥MYR 2000	567	14.3
Living Arrangements		
Staying alone	295	7.4
Staying with partner/family	3682	92.6

**Table 2 ijerph-20-04342-t002:** Prevalence of poor self-rated health among Malaysian older persons aged 60 years above (n =1438).

Variable	Poor Self-Rated Health			*p*-Value
	Unweighted Count	Prevalence	95% CI	
Gender				0.164
Male	679	31.2	27.61–35.10	
Female	759	34.0	29.23–39.05	
Locality				0.077
Urban	543	30.8	26.11–36.03	
Rural	895	37.5	32.25–42.98	
Education Level				<0.001
Non-formal	359	42.8	37.59–48.17	
Primary	732	36.7	31.55–42.13	
Secondary	281	26.3	21.04–32.28	
Tertiary	66	20.3	14.27–27.94	
Employment Status				<0.001
Employed	328	26.7	21.84–32.27	
Unemployed	1110	34.5	30.63–38.66	
Marital Status				<0.001
Single	538	38.0	33.02–43.15	
Married	900	30.1	26.31–34.27	
Individual Monthly Income				<0.001
<MYR 1000	1011	37.9	33.33–42.71	
MYR 1000–MYR 1999	280	30.1	25.85–34.67	
≥MYR 2000	137	20.9	16.00–26.73	
Living Arrangements				0.806
Staying alone	109	33.7	24.26–44.70	
Staying with partner/family	1329	32.6	28.83–36.53	
Self-Reported Diabetes				0.005
Yes	422	37.9	32.56–43.63	
No	1016	30.7	26.76–34.85	
Self-Reported Hypertension				<0.001
Yes	837	37.0	32.44–41.88	
No	601	28.1	24.49–31.98	
Self-Reported Hypercholesterolemia				0.003
Yes	633	37.1	31.80–42.71	
No	805	29.5	25.72–33.49	
Current Smoker				0.805
Yes	232	32.0	26.36–38.24	
No	1206	32.7	28.77–36.99	
Physical Activity				<0.001
Active	813	27.1	23.03–31.56	
Inactive	624	45.8	40.18–51.43	
Depressive Symptoms				<0.001
Yes	301	62.7	55.56–69.24	
No	1019	27.6	23.81–31.76	
Consumption of Fruits and Vegetables				0.472
<5 serving/day	1351	32.4	28.55–36.40	
≥5 serving/day	78	35.6	26.53–45.94	
Limitations in ADLs				<0.001
Present	387	55.8	48.82–62.62	
Absent	1050	27.9	24.19–31.97	

**Table 3 ijerph-20-04342-t003:** Univariate and multivariate logistic regression analysis for poor self-rated health among Malaysian older persons aged 60 years above (n = 1438).

Variable	Crude OR (95% CI)	*p*-Value	Adjusted OR (95% CI)	*p*-Value
Gender				
Male	1		1	
Female	1.13 (0.95, 1.35)	0.164	0.83 (0.64, 1.07)	0.148
Locality				
Urban	1		1	
Rural	1.34 (0.97, 1.86)	0.077	1.06 (0.75, 1.49)	0.741
Education Level				
Non-formal	2.95 (1.88, 4.61)	<0.001	1.46 (0.90, 2.39)	0.126
Primary	2.28 (1.46, 3.56)	<0.001	1.56 (0.98, 2.50)	0.063
Secondary	1.40 (0.91, 2.16)	0.124	1.22 (0.80, 1.88)	0.370
Tertiary	1		1	
Employment Status				
Employed	1		1	
Unemployed	1.45 (1.17, 1.79)	0.001	0.95 (0.74, 1.23)	0.700
Marital Status				
Single	1.42 (1.18, 1.71)	<0.001	1.11 (0.89, 1.37)	0.360
Married	1		1	
Individual Monthly Income				
<MYR 1000	2.32 (1.72, 3.12)	<0.001	1.66 (1.22, 2.26)	0.001
MYR 1000-MYR 1999	1.63 (1.18, 2.25)	0.003	1.34 (0.99, 1.80)	0.059
≥MYR 2000	1		1	
Living Arrangements				
Staying alone	1.05 (0.69, 1.61)	0.806	0.83 (0.51, 1.36)	0.457
Staying with partner/family	1		1	
Self-Reported Diabetes				
Yes	1.38 (1.11, 1.73)	0.005	1.15 (0.94, 1.42)	0.168
No	1		1	
Self-Reported Hypertension				
Yes	1.51 (1.28, 1.78)	<0.001	1.23 (1.02, 1.49)	0.032
No	1		1	
Self-Reported Hypercholesterolemia				
Yes	1.41 (1.13, 1.77)	0.003	1.22 (0.97, 1.54)	0.087
No	1		1	
Current Smoker				
Yes	0.97 (0.74, 1.27)	0.805	0.86 (0.62, 1.21)	0.387
No	1		1	
Physical Activity				
Active	1		1	
Inactive	2.27 (1.74, 2.96)	<0.001	1.40 (1.08, 1.82)	0.011
Depressive Symptoms				
Yes	4.40 (3.09, 6.26)	<0.001	2.92 (2.01, 4.24)	<0.001
No	1		1	
Consumption of Fruits and Vegetables				
<5 serving/day	0.86 (0.58, 1.29)	0.472	0.75 (0.51, 1.09)	0.130
≥5 serving/day	1		1	
Limitations in ADLs				
Present	3.26 (2.41, 4.42)	<0.001	1.82 (1.31, 2.54)	<0.001
Absent	1		1	

## Data Availability

The datasets used in this study are not publicly available.

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
