# Peer review of "Poor Self-Rated Health and Associated Factors among Older Persons in Malaysia: A Population-Based Study"

_ijerph, 2023, doi:10.3390/ijerph20054342_

Round 1
Reviewer 1 Report
Please see the attached file.

Reviewer 2 Report
Clearly written and clinically relevant. The study correctly discussed the expected findings of disability, low income and depression to be associated with poor SRH.
However, hypertension is not a common cause of poor SRH, which needs further discussion about possible reasons. In addition, diabetes mellitus, which is expected to have a negative impact on SRH than hypertension was surprisingly not. Please discuss.
Minor typos to improve readability of manuscript:
Line 49, reported had, please change to: reported to have
Line 63, NCD, please spell out
Line 78, to ensured, please change to: ensure
Line 134/135 adequate consumption (5 servings per day) and inadequate consumption (5 servings per day), please correct, 5 servings should not be the same adequate and inadequate.
Line 174, they with less educated, please change to: those less educated
Line 253, should benefited, please change to: benefit
The manuscript explores the self-rated health (SRH) of Nationwide Malaysian population and its relation to underlying health conditions. The topic is clinically relevant and not investigated before in Malaysian population. The topic is original. Not been investigated before in Malaysian population. Manuscript is well written.Clear, well-written and easy to read.
Round 2
Reviewer 1 Report
The paper has been now much improved.